# "Vulnerabilities and compound risks of escalating climate disasters in the Brazilian Amazon"

Patrícia F. Pinho [1] ✉, Rafaella Silvestrini[1], Martha Fellows[1], Letícia Perez [2], Ane Alencar [1], Carolina Guyot[1], Paulo Moutinho [1], David Lapola [3] & Lindsay Stringer [4]

The Brazilian Amazon is severely impacted by extreme climate events, with 1.8 million people (6.4% of the Brazilian Amazon's population) affected by climate-related disasters each year between 2018 and 2022. Yet, how climate disasters specifically affect human populations in different municipalities in the Amazon is underexplored. We target this gap, presenting a region-wide spatiotemporal assessment of climate-related disasters and social vulnerability across the Brazilian Amazon over the period 2000–2022, considering floods, droughts, heatwaves, fires, and landslides, and using a compound risk lens. Analysis of secondary data shows disaster frequency surged, with wet events rising five-fold, fires tenfold, and droughts and heatwaves tripling. Economic losses rose 370% reaching USD 634.2 million annually, with wet events most damaging. Farming sustained over 60% of total losses, followed by infrastructure, housing, and health services. Smaller municipalities, which host 61% of the region's Indigenous population, experienced the highest relative impacts, including a 9.58% loss in economic growth and lower Social Progress Index scores. Data on non-economic losses and damages were lacking, but further exacerbated the impacts in these vulnerable areas. Findings underscore that climate change is a poverty multiplier, and highlight the urgent need for adaptation policy interventions to be justice-centered.

In the globally important Brazilian Amazon, comprehending the confluence of risks, exposure, and vulnerabilities is central to alleviating human experiences in the face of climate change disasters[1]. This region, the world's largest tract of tropical forest, is central to global climate stability[2,3], with high biological and cultural diversity[4]. Its collapse is one of the major ecological tipping points under increasing global temperatures and greenhouse gas emissions[5]. For over two decades, the Brazilian Amazon has been severely impacted by extreme droughts and heatwaves, floods, and fires of intense magnitude and

frequency[6,7]. These events have often occurred consecutively, sometimes overlapping, posing substantial challenges to both the local people, economy and institutional adaptation efforts[8,9].

Global climate disaster data already underscore the acute impacts faced by hundreds of millions in the Global South, who confront substantial challenges in preventing climate change related loss and damage and protecting livelihoods, health, and food security, often compounded by limited adaptive capacities[10–12]. However, the impacts on the health, food security, livelihoods and cultural heritage of

[1]Instituto de Pesquisa Ambiental da Amazônia (IPAM, Amazon Environmental Research Institute), Brasília, Federal District, Brazil. [2]Centro de Estudos do Agronegócio (FGV Agro, Agribusiness Studies Center – Fundação Getulio Vargas), São Paulo, São Paulo, Brazil. [3]Centro de Pesquisas Meteorológicas e Climáticas Aplicadas à Agricultura (CEPAGRI), Universidade Estadual de Campinas (UNICAMP), Campinas, São Paulo, Brazil. [4]Department of Environment & Geography, University of York, York, UK. ✉e-mail: patricia.pinho@ipam.org.br

Amazon people, especially Indigenous groups, are profound, irreversible but still poorly understood[13,14]. Research on unquantifiable, non-economic losses and damages (NELD), including values, social norms and culture, to which damage is irreversible[15] has demonstrated a devastating impact on these groups, particularly through the erosion of livelihoods, ecological knowledge, and place-of-birth attachment, alongside increasing migration[16]. Disasters are also affecting children's biological and educational development causing intergenerational disadvantages within Brazilian society[17], while migration is affecting Indigenous people's emotional health through perceived depressive symptoms, distress and suicides[18]. These exposures and vulnerabilities arise not only from poverty but also from various dimensions of inequality, including ethnicity, which distinctly shapes local experiences of climate disasters[19]. Despite the critical need emerging from the little existing research available, there remains an urgent gap in understanding and data on how climate disasters specifically affect human populations, their livelihoods, and regional economies in the Brazilian Amazon[10,13].

This paper advances understanding of social vulnerability and exposure to climate disasters, specifically in the Brazilian Amazon by employing a compound risk perspective[19] to comprehensively assess socioeconomic impacts and underlying vulnerabilities. We delineate the spatiotemporal patterns of climate-related disasters[20]—including floods, landslides, droughts, heatwaves and fires—over a period of twenty-two years (2000–2022) and conduct a municipal-level analysis to identify the demographics most affected, quantifying both economic and non-economic losses and damages. Our findings highlight the uneven burden of climate disasters, with smaller municipalities and Indigenous populations bearing the brunt of the impacts, underscoring the importance of a justice framing in managing climate impacts. Our comprehensive assessment provides an important foundation for developing effective policy interventions in critical systems like the Brazilian Amazon that integrate considerations of justice with economic development, human rights, and climate risks, offering actionable insights for policymakers committed to enhancing resilience. Without addressing exposure, vulnerability, and the limits to adaptation among other structural constraints faced by Brazilian Amazonian municipalities and Indigenous populations, compound risks beyond the 1.5 °C warming target set by the Paris Agreement will be increasingly challenging for adaptation[11].

## Results

### Shifting climate disasters

Climate-related disasters in the Brazilian Amazon have shown a pronounced increase and shift in prevalence over the two-decade period in this study. Initially dominated by wet events (floods and intense rainfall), the region has seen a substantial increase in droughts, heatwaves, and fires since 2005 (Fig. 1A, B). These disasters, defined as severe disruptions caused by hazardous events interacting with population exposure, vulnerability, and capacity, have led to extensive human, material, economic, and ecological losses[21]. The frequency of wet events increased by 124% from the mid-2000s (2006–2010) to 2018–2022. When comparing the mean annual values from 2000–2005 to 2018–2022, wet events have risen more than fivefold, and landslides increased by 189% over the same mid-2000s period, showing an eightfold surge in comparison to 2000–2005. Similarly, drought and heatwave occurrences have tripled when comparing 2000–2005 to 2018–2022, with a 15% increase noted when comparing 2006–2010 to 2018–2022. Fire events, in particular, have surged by 409% compared to the mid-2000s, resulting in a more than tenfold increase from 2000–2005 to 2018–2022 (Fig. 1B). Other climatic disasters also exhibited substantial increases over these periods.

Spatial analysis indicates that while wet events are uniformly distributed across the municipalities of the Brazilian Amazon, droughts predominantly affect the northern (RR), central (AM), and southern and southeastern municipalities (MT, TO, and MA) (Fig. 1A). Fire disasters are primarily concentrated in the municipalities of RR, MT, AC, and TO. Conversely, landslides and other disaster types do not exhibit a clear pattern of distribution across the region.

## Disaster exposures on the rise: a growing concern for people

Our analysis from 2018 to 2022 reveals a sharp increase in the number of individuals affected by disasters, with 1.78 million people, or 6.44% of the region's population, now exposed annually. This marks a dramatic rise from the 2000–2004 period, when only 2316 people, or 0.11% of the population, were exposed (Fig. 2A). Compared with the 2006–2010 period, 541,180 people, or 2% of the population, were exposed annually.

Wet events accounted for the majority of this exposure, affecting 62.08% of the population, followed by droughts and heatwaves (13.65%), fires (11.73%), landslides (8.7%), and other events (3.84%) (Fig. 2A, B). In years with extreme conditions, wet disasters impacted 408 municipalities, accounting for 53% of the region, and were evenly distributed throughout the Region (Fig. 2B). Among these, more than 75% of the population was affected in 60 municipalities. Severe droughts, heatwaves, and fire events affected 130 municipalities, about 17% of the region, with these primarily concentrated in the states of Acre (AC), Mato Grosso (MT), the southwest of Pará (PA), and Roraima (RR) (Fig. 2B).

## Climate disasters cause substantial economic impacts

Economic losses from climate-related disasters in the Brazilian Amazon reveal a striking increase over two decades. From 2000 to 2022, total financial damages were substantial, totaling approximately USD 5.78 billion. Annual mean losses have surged from USD 132.8 million during 2006–2010 to USD 634.2 million in the period 2018–2022, marking an increase of 4.7 times (Fig. 3A). Breaking down the losses by disaster type, wet events were the most costly, accounting for 84.78% of total losses (USD 4.9 billion). Droughts and heatwaves followed, incurring 12.1% of the losses (USD 700 million), with landslides, fires, and other climate disasters contributing less substantially (Fig. 3B). Within wet disasters, losses were particularly in farming (agriculture and livestock), and amounted to USD 2.77 billion (57% of losses from wet events). Infrastructure and housing were also heavily affected, with damages of USD 962 million and USD 447 million, respectively. Health-related impacts, including damage to healthcare facilities and services, totaled USD 53 million. Droughts and heatwave disasters primarily impacted farming, accounting for USD 650.5 million (97% of losses from these events). Health-related costs amounted to USD 5.87 million. Likewise, fire disasters resulted in USD 64.2 million in losses, predominantly affecting farming (92%), with health-related losses totaling USD 0.58 million. Landslide disasters resulted in total losses of USD 101 million, with 45% (USD 46.5 million) attributed to infrastructure damages and 18% (USD 18.9 million) to housing (Fig. 3A). Across all disaster types, farming sustained 60.3% of total losses (USD 3.5 billion). Damages to public infrastructure and housing were also notable, totaling USD 1.1 billion (17%) and USD 470 billion (8%), with health-related costs of USD 61.5 million (1%).

## Small municipalities and indigenous peoples are disproportionately impacted

Our analysis reveals a striking disparity in the impact of climate-related disasters across different-sized municipalities and Indigenous populations within the Brazilian Amazon. The smallest municipalities, defined as those with populations under 50,000, suffered the most severe economic consequences of climate disasters, with losses averaging 9.58% of their economic growth from 2002 to 2020. In contrast, municipalities with populations exceeding 50,000 and 300,000 saw

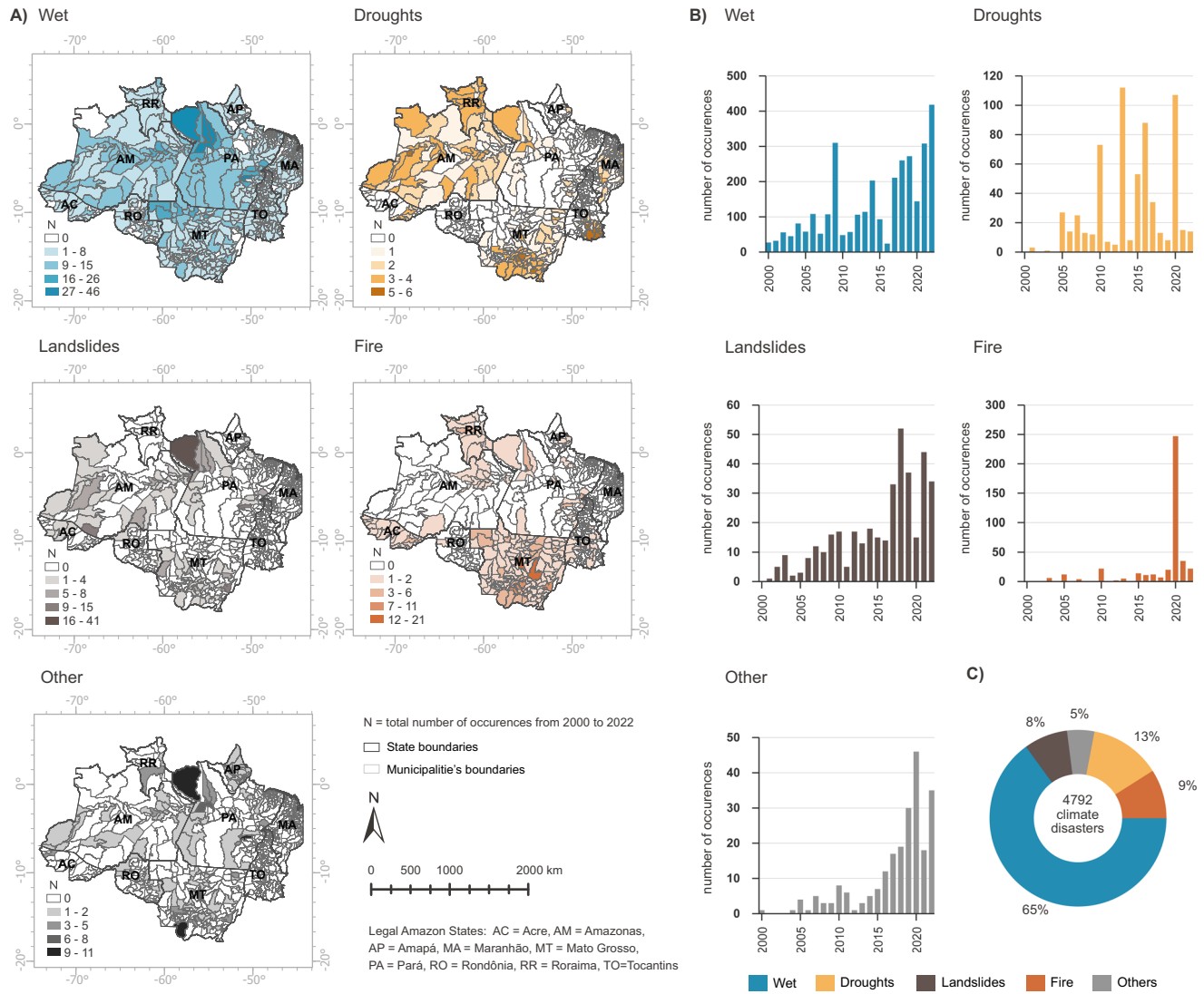

**Fig. 1 | Number of climate disasters over municipalities in the Brazilian Amazon from 2000–2022. A** Spatial distribution of total number of disasters by municipality and disaster typology from 2000 to 2022; **B** Temporal distribution of disasters over the study years in all municipalities. **C** Total number of disasters by climate typology from 2000–2022. Data Source: Brazilian Atlas of Disasters. Graphic and map design by Rafaella Almeida Silvestrini, using Microsoft Excel (Microsoft 365) and ArcGIS Pro v3.1.3, respectively. Post-editing by Carolina Guyot in Adobe Illustrator v 28.7.7.Basemap sources: state and municipality boundaries from the Brazilian Institute of Geography and Statistics (IBGE), licensed under a Creative Commons Attribution 4.0 International License (CC BY 4.0, https://creativecommons.org/licenses/by/4.0/).

economic losses constituting 3.8% and 1.3% of their economic growth, respectively (Fig. 4D).

Municipal population size can be considered as a vulnerability, because in Brazil, municipalities with less than 20,000 people are not required to have a city master plan to guide the city's development. Often master plans indicate risk areas that should not be occupied. Smaller municipalities also show consistently higher numbers of climate disasters and lower Social Progress Index (SPI) scores compared to larger cities, while they also host a greater proportion of Indigenous residents (Fig. 4A–C; see Methods for details on SPI). Statistical analysis indicates significant correlations between smaller municipalities and the number of Indigenous inhabitants, with a Kendall-tau of -0.063 and Spearman of 0.025 for SPI, and a Kendall-tau of 0.106 and Spearman of 0.16 for the proportion of Indigenous populations.

These findings underscore the vulnerability of smaller municipalities predominantly inhabited by Indigenous peoples to climate disasters and highlight the urgent need for targeted adaptation and resilience strategies to reduce these risks for these populations.

## Compound risks across the Brazilian Amazon

Our study highlights pronounced variations in the multidimensional components of exposure and vulnerability to climate risks across municipalities in the Brazilian Amazon. We have identified distinct levels of compound risk based on the exposure of people and assets and the resultant economic impact. At the highest level of risk, defined by a "red rectangle," nine municipalities, each with a population under 20,000 people, experienced the most severe effects (Fig. 5A, B). In these areas, disasters affected over 50% of the population, and total economic losses exceeded 50% of their economic growth over the period 2002–2020 (Fig. 5A). This category includes municipalities located across various states: four in Amazonas (AM), two in Maranhão (MA), and one each in Mato Grosso (MT), Pará (PA), and Tocantins (TO) (Fig. 5A, B). The "orange rectangle" represents the second highest level of risk, comprising 187 municipalities (24.2% of the region) where disasters affected more than 50% of the population, yet the economic losses were less than 50% of the economic growth. This group is geographically diverse, with 119 municipalities (15% of the total) having populations under 20,000 (Fig. 5A, B). The "green rectangle" indicates

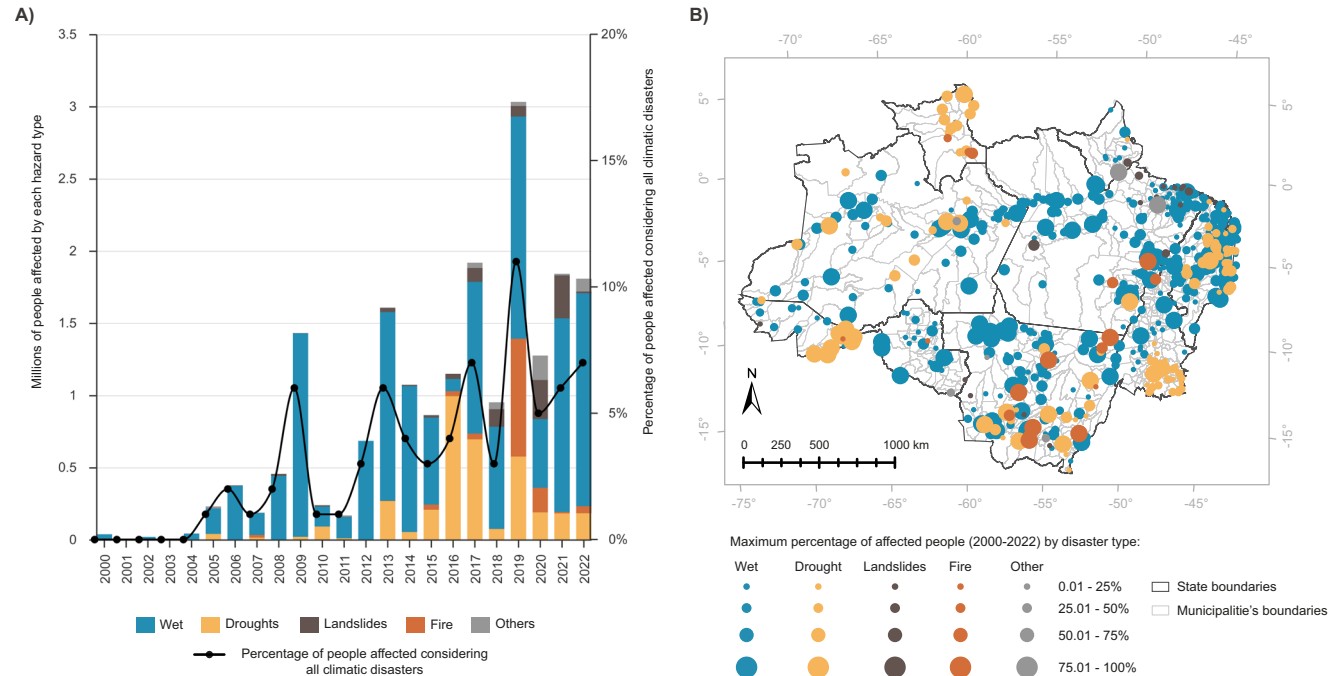

**Fig. 2 | Population exposed to disasters in the municipalities of the Brazilian Amazon over the period 2000–20022. A** Temporal distribution of number of people affected by disaster typology (millions/year) and percentage of population exposed considering all disasters; **B** Maximum annual percentage of affected people between 2000–2022 and its corresponding disaster typology by municipality. Graphic and map design by Rafaella Almeida Silvestrini, using Microsoft

Excel (Microsoft 365) and ArcGIS Pro v.3.1.3, respectively. Post-editing by Carolina Guyot in Adobe Illustrator v 28.7.7.Basemap sources: state and municipality boundaries from the Brazilian Institute of Geography and Statistics (IBGE), licensed under a Creative Commons Attribution 4.0 International License (CC BY 4.0, https://creativecommons.org/licenses/by/4.0/).

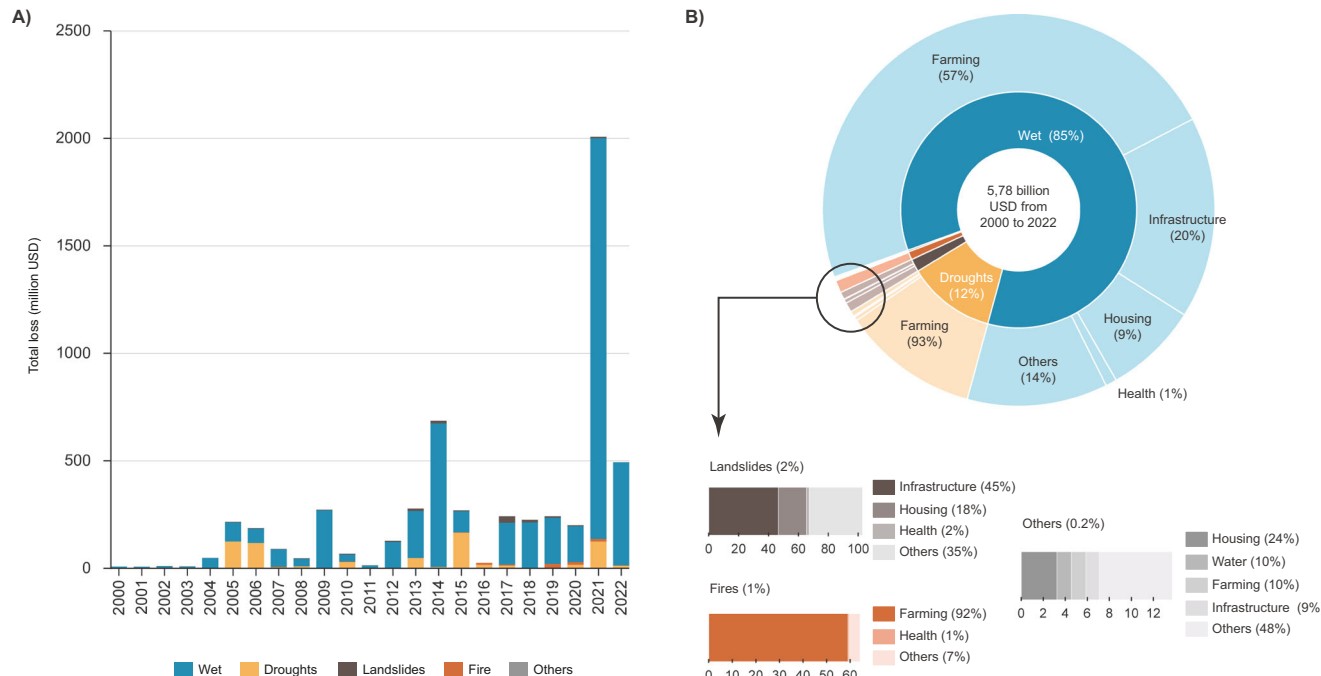

**Fig. 3 | Asset losses due to climate disasters in the Brazilian Amazon. A** Annual economic losses by disaster type. **B** Total losses from 2000 to 2022, categorized by disaster type and affected sectors. Data source: Brazilian Digital Atlas of Disasters.

Graphic design by Rafaella Almeida Silvestrini, using Microsoft Excel (Microsoft 365). Post-editing by Carolina Guyot in Adobe Illustrator v 28.7.7.

a medium level of risk and includes just two municipalities located in eastern Amazonas, along the Amazon River, with populations under 50,000. Here, the climate disasters affected less than 50% of the population, but the economic losses totaled more than 50% of their

growth between 2002 and 2020. The lowest level of risk, defined by a "blue rectangle," includes 575 municipalities (74.5% of the region). In these areas, climate disasters affected less than 50% of the population, and the economic losses were less than 50% of their economic growth.

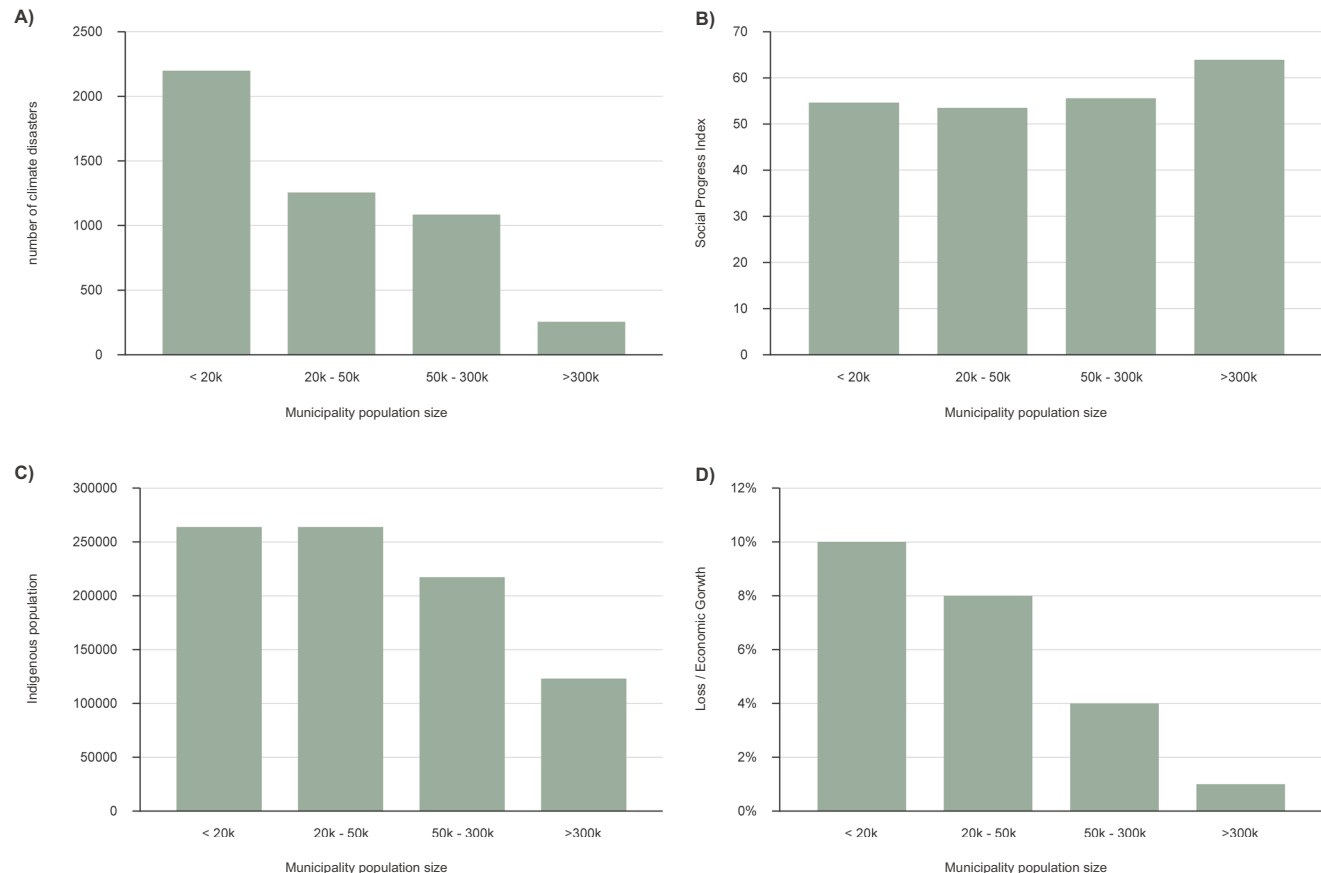

**Fig. 4 | Vulnerability to climate disasters based on municipalities' population size.** Vulnerability is represented on the X-axis by population size and on the Y-axis as follows: **A** number of climate disasters; **B** Social Progress Index; **C** Indigenous population; and **D** Loss to Economic Growth ratio. Graphic design by Rafaella Almeida Silvestrini, using Microsoft Excel (Microsoft 365). Post-editing by Carolina Guyot in Adobe Illustrator v 28.7.7.

A majority, 355 municipalities (50%), have populations less than 20,000 and are spread across the Brazilian Amazon.

## Discussion

Over the 22 year study period, our findings reveal a substantial increase in the frequency and compounding impacts of climate-related disasters across the Brazilian Amazon. Nearly 2 million people are now affected annually by floods, droughts, heatwaves, fires, and landslides. Financial losses have escalated more than fourfold since the early 2000s, with floods affecting the greatest number of affected people and inflicting most economic damage, particularly across agriculture, livestock, infrastructure, housing, and health services. Likewise, droughts, fires, and heatwaves are intensifying social vulnerability by isolating communities, disrupting access to education and healthcare, increasing food and fuel insecurity, worsening health conditions, and driving conflict[22]. The historical pattern of disasters in the Brazilian Amazon has shifted from predominantly wet events to an increasing frequency of droughts, heatwaves, and fires, with recurrent droughts occurring earlier than anticipated[6,23,24]. Fires in specific regions show greater recurrence, with return intervals far shorter than ecological thresholds[24]. Although improved reporting may contribute marginally, robust evidence of increasingly frequent and severe droughts, floods, and fires confirms that the observed trends reflect a real intensification of climate extremes in the Amazon[4,21,22]. These hazards profoundly impact the region's climate-sensitive livelihoods, such as agriculture, livestock, fisheries, and extractivism, as well as the already precarious basic services and infrastructure[8,13,14,25,26]. Indigenous people and local communities, dependent on natural resources and river systems are particularly vulnerable to the escalating severity and frequency of

these climate disruptions[10,14,27–29]. The spatiotemporal exposure of municipalities and populations in the Brazilian Amazon is closely linked to social vulnerability distribution, including low social development, particularly in terms of water, sanitation, health and education, indicating critically low capacities to meet basic needs[11,12,24]. These areas also coincide with the highest proportion of Indigenous inhabitants, highlighting a critical overlap between social inequalities, Indigenous populations and climate disasters[10]. The intersections of ethnicity and race, as well as age, gender, socioeconomic class, shape the climate risks and differential impacts experienced by these groups[30]. The historical and ongoing colonial processes that create social, economic, and political inequalities, and marginalize Indigenous people and their livelihoods, drive the unevenness of opportunities to manage climate disasters[31]. Most municipalities in the Brazilian Amazon (~61%) originated from Indigenous settlements emerging from the colonial exploitation of natural resources in their territories[32]. This colonial dominance has persisted and is a key driver of unequal climate impacts in the Global South more broadly[31].

Brazilian Amazon municipalities are detached from centers of power, resources and decision-making, leading to ill-preparedness to plan for and respond to floods, droughts and their impacts. A large number of municipalities are located up to 2820 km from state capitals and rely directly on major river systems to meet essential needs, compromising timely responses during extreme events[33]. Livelihoods in municipalities with larger Indigenous populations are often informal, invisible, and do not benefit from information sharing linked to hazard monitoring for risk reduction[34]. The official data of the National Center for Monitoring and Early Warning of Natural Disasters (Cemaden) monitors the Brazilian Amazon's larger urban centers in relation

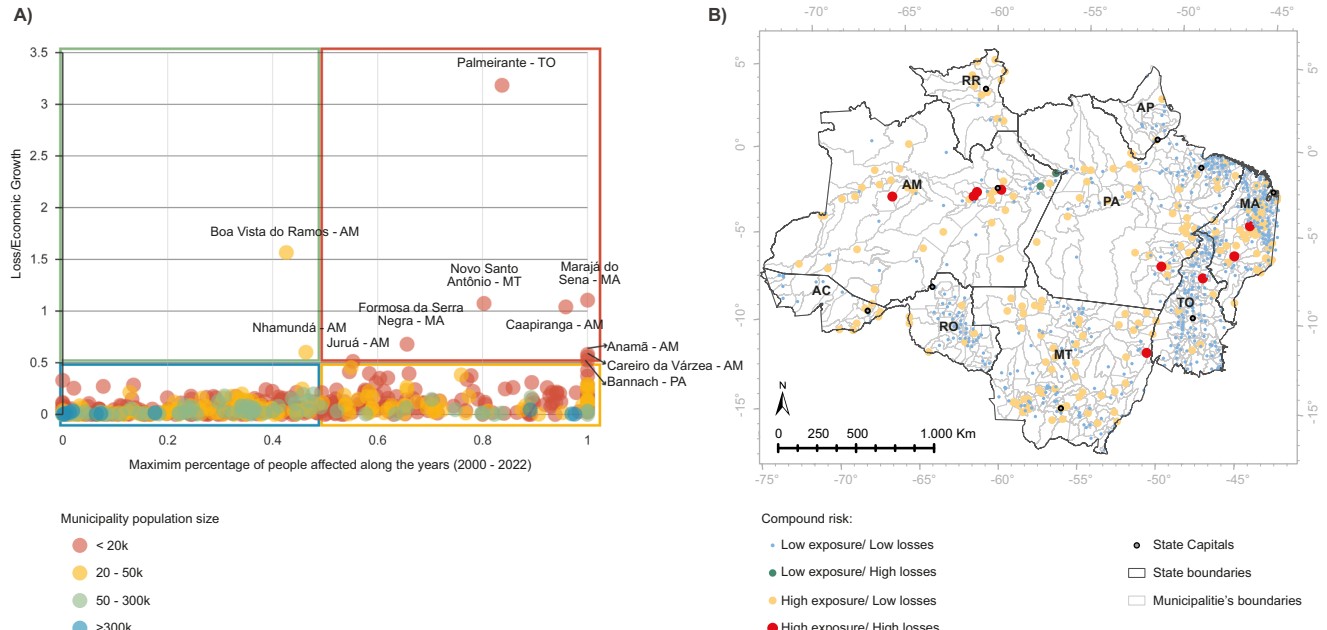

**Fig. 5 | Compound risk in each municipality, analyzing both population exposure in the year of the most extreme disaster and total economic loss in relation to economic growth. A** Scatterplot classifies municipalities into 4 different levels of risk: red (high/high), grouping municipalities in which more than 50% of the population was affected in the most extreme disaster year and in which total economic losses surpassed 50% of municipality economic growth; yellow (high/low), grouping municipalities in which more than 50% of population was affected in the most extreme disaster year, but total economic losses were less than 50% of municipality economic growth; green (low/high), comprising municipalities with less than 50% people affected in the most extreme year and with total losses of more than 50% of economic growth; blue (low/low), comprising municipalities with less than 50% people affected in the most extreme year and with total losses of less than 50% of economic growth. **B** map showing the spatial distribution of compound risk (classification obtained in **A**) Graphic and map design by Rafaella Almeida Silvestrini, using Microsoft Excel (Microsoft 365) and ArcGIS Pro v3.1.3, respectively. Post-editing by Carolina Guyot in Adobe Illustrator v 28.7.7.Basemap sources: state and municipality boundaries from the Brazilian Institute of Geography and Statistics (IBGE), licensed under a Creative Commons Attribution 4.0 International License (CC BY 4.0, https://creativecommons.org/licenses/by/4.0/).

to floods and landslides, but it does not monitor droughts and heatwaves that affect mostly vulnerable municipalities and people[35]. Thus, data on smaller municipalities are largely unavailable[36] reinforcing vulnerability in these areas. Globally, Indigenous peoples and forest-dwelling communities in urban centers are often missing from national poverty assessments, and frequently excluded from planning, including adaptation planning, despite potentially representing a small fraction of the population[35,36]. It is highly worrisome that these vulnerable Indigenous municipalities are unable to establish prevention, preparedness, response and recovery actions given data, financial and human capital constraints[37].

Although Indigenous populations and traditional communities may have traditional environmental planning systems rooted in local knowledge, the absence of formal municipal master plans excludes their priorities from national surveys and disaster policy frameworks, making it more difficult to implement adaptation measures[38,39]. Yet the National Civil Defense and Protection Policy (PNPDEC) attributes the responsibility to the municipalities to map risk areas, identify threats, evaluate susceptibilities, vulnerabilities, and plan for disaster risk relief and relocation[32,34].

Higher governance levels continue to impose colonial conditions on relatively poor municipalities in the Brazilian Amazon, limiting their access to financial relief and assistance to affected populations[34]. Meanwhile, disasters might be driving irreversible losses among historically marginalized Indigenous, and traditional populations whose livelihoods and vulnerabilities remain largely invisible in national metrics. Consequently, losses experienced by the Indigenous people and their livelihoods in affected municipalities are likely to be more severe than is shown by the data. This aligns with broader evidence that climate change acts as a "poverty multiplier," disproportionately affecting those with the least capacity to adapt, while their losses, often

related to non-market livelihoods and basic services remain largely unaccounted for in macroeconomic metrics[40].

Compound risks in the Brazilian Amazon are causing losses to increase over time, leading to the silent impoverishment of people and municipalities, and increasing the adaptation gap. These findings underscore how consecutive climate disasters are deepening vulnerability in the Brazilian Amazon, as repeated shocks over multiple years reduce recovery time, erode resilience, and create chronic disruption in already marginalized municipalities. Recent global evidence confirms that climate shocks disproportionately impact low-income populations, who are nearly twice as likely to suffer adverse effects compared to higher-income groups[10,41]. Over 90% of studies reviewed[42] report that droughts, floods, and extreme heat disproportionately harm low-income populations. These effects persist across life stages, eroding health, education, and labor outcomes, and are often intensified by structural inequalities such as concentration in high-risk zones[42] and dependence on climate-sensitive livelihoods like agriculture[40]. These findings align with our results in the Brazilian Amazon, where small and vulnerable municipalities, often home to Indigenous and traditional populations, experience repeated and compounding shocks that deepen existing poverty and social fragility. Addressing these multifaceted challenges requires a justice lens[43], integrating Indigenous and local knowledge into attribution, vulnerability, and adaptation science[44,45]. By incorporating Indigenous and local knowledge, we can better assess vulnerabilities, capacities, and losses from extreme events, enhancing monitoring and early warning systems, and ensuring equitable access to resources, recovery, rehabilitation, and compensation for inclusive and effective adaptation[46]. At the same time, careful and fair relocation of the affected population is necessary to avoid NELD due to breakdown of social networks, livelihoods and cultural practices. Yet, NELD evaluation and policy

mechanisms remain an under-assessed area of academic investigation and policy practice in the Brazilian Amazon and Global South[47,48]. Advancing this agenda requires more granular data to capture how overlapping shocks affect climate-sensitive livelihoods—especially among Indigenous peoples—and supporting just and resilient policy responses.

Inter-related forms of invisibility and social and spatial marginalization, neglected multi-dimensional social development and uneven climatic monitoring and forecasting, urgently demand that a justice lens be placed on climate related disasters in the Brazilian Amazon. Justice enables us to draw attention to structural inequalities to reorient efforts to reduce risks[47], in a context in which our data has shown smaller municipalities and Indigenous groups who are highly and directly dependent on natural resources are bearing the greater burden of exposure and vulnerability when climate disasters hit. Justice in the Brazilian Amazon needs to address the historical colonial legacy of exclusion, alongside unequal benefits and access to resources, opportunities, socioeconomic support, and wealth, that hinder their agency and adaptive capacity[49]. At-risk municipalities, the government and private sector are accountable and need to provide what is absent (e.g. accessible transportation, access to health systems, water, and food supplies, schools, and social support) including early warning systems, presenting a need for distributive justice[11]. Justice actions need to permeate the private sector through socioecological and climate related safeguards, primarily linked to activities exploiting the Brazilian Amazon's natural resources, lowering near-term climate and social risks. The region also needs a similar and urgent inclusion into article 8 of the Paris Agreement establishing the mechanisms by which Annex I countries should support adaptation finance to build resilience and reduce loss and damage to affected populations in the Global South. Climate change impacts in the Brazilian Amazon are unfolding faster than previously anticipated, while responses remain highly inefficient and spatially fragmented, highlighting the urgent need for legal and integrated mechanisms to avert further losses and damages. Looking ahead, under future climate change scenarios and in the absence of effective and urgent mitigation and adaptation the intensification of climate disasters in the Amazon is projected to worsen with indigenous people and traditional communities facing a growing risk burden compromising sustainable development[10]. Recent modeling efforts show that many municipalities, particularly smaller and vulnerable, will face compounding disaster risks over two consecutive 25-year intervals (2015–2039 and 2040–2064), with risks accelerating under high-emissions scenarios[50]. These projections show that the frequency and severity of extreme events such as floods, droughts, and heatwaves will escalate across Amazonian subregions, exacerbating existing vulnerabilities and potentially pushing population and economies beyond recovery thresholds. This underscores the urgency of anticipating compound climate risks and advancing inclusive, equity-driven resilience policies across the region.

## Methods

The evaluation undertaken here was elaborated in accordance with the disaster risk components outlined in the IPCC's AR6 report[5,20]. "Hazards" focus on the identification and quantification, in space and time, of climatic extreme events or disturbances directly related to them, such as fires and landslides, in the Brazilian Amazon region. "Exposure" analyzed the population, infrastructure, housing, agriculture and public services exposed to the hazards. "Vulnerability," assessed the social and demographic conditions of the affected population and municipalities, including the financial capacity of municipalities to respond to disasters. "Compound risk" was obtained by analyzing the joint impact of the three previous risk components, allowing the identification of the municipalities that faced the highest or lowest disaster-related risks. All analyses were conducted at the municipality level for the Brazilian Amazon for the period 2000–2022.

The following sections describe the methods and datasets employed for each disaster risk component.

### Hazards

Climatic extreme events data were obtained from the Brazilian Atlas of Disasters[51] which documented 5010 disaster events between 2000 and 2022 in the Brazilian Amazon, with 4792 of these occurrences being related to climatic extremes. The Atlas classifies disasters into sixteen categories, eleven of which are associated with climate-related hazards, such as different types of floods, droughts, and extreme heat conditions. In this study, these eleven original categories were consolidated into five major hazard types relevant to the Brazilian Amazon: (1) Wet (different kinds of floods and intense rain); (2) Drought and Heatwaves; (3) Fire; (4) Landslides; (5) Other climate-related extremes (gales, tornadoes, cold waves and haze).

The total annual number of climate-related hazards reported for the Brazilian Amazon was computed between 2000 and 2022. We used 5-year intervals (2000–2005, 2006–2010, 2018–2022) to capture medium-term trends while minimizing annual variability. This resolution enhances interpretability, and consistent year-on-year increases suggest the main patterns are robust across different temporal aggregations. Subsequently, the spatial and temporal distributions of these observed hazards were examined. The spatial distribution was established by mapping the total number of occurrences (2000–2022) for each of the 772 municipalities. The temporal distribution was evaluated by analyzing the annual frequency of climate disasters across the entire Brazilian Amazon for each of the five hazard types.

### Exposure

Exposure was evaluated according to two numerical indicators: human population and asset exposure. Human population exposure, we used the number of people affected by disasters annually, available from the Brazilian Atlas of disasters, to assess population exposure. The yearly total population data per municipality was obtained from the IBGE (2023)[52,53] and is based on decennial census counts (years 2000, 2010, and 2022) or on population size estimates (years 2001-2009; 2011-2021). We calculated the proportion of the population affected by disasters by dividing the annual number of affected people by the total population of each municipality for the corresponding year and disaster type. To observe the temporal distribution of population exposure, we plotted the number of people affected by hazard type and the percentage of population affected yearly. This allowed us to identify, for each municipality, the disaster type that affected the highest percentage of population in the study years, with the resultant map showing the type with highest population exposure across all years.

Asset exposure data were sourced from the Brazilian Atlas of Disasters, which documents financial losses associated with each climate-related disaster. Losses resulting from disasters are classified into several categories, including private and public infrastructure damage (reclassified into health, education, community centers, construction works, housing and other) and private and public losses (divided into farming, industries, and services, including water supply and public health assistance). For each disaster type, losses pertaining to the above-mentioned assets categories were gathered annually by municipality for the period 2000–2022. Monetary values were made available originally in Brazilian currency (Reais), updated to monetary values for 2022, and were converted into 2022 dollars using the conversion rate of 1 US dollar = 0.19 reais. In addition to presenting the annual distribution of losses over time and by hazard type, we also demonstrated the allocation of total losses among asset categories for each hazard type.

### Vulnerability

The vulnerability component was composed of social and demographic characteristics and the response capacity of the municipalities.

For Social and demographic characteristics, we analyzed characteristics of the population using the Social Progress Index (SPI)[54] and the Indigenous population within each municipality[52]. The SPI comprehensively measures the social and environmental performance of territories by integrating globally relevant human development indicators alongside indicators adapted for the context of the study region, encompassing aspects like maternal mortality rate, access to water, school enrollment, deforestation rate, early pregnancy in childhood and adolescence, and violence. The Indigenous population was measured in the last census in 2022 by IBGE (2023)[52]. We calculated the percentage of Indigenous people in relation to the total municipality population and correlated population exposure (as estimated in section 2.1) with the SPI and the percentage of Indigenous people. We used the non-parametric correlation indices of Spearman ranking correlation and Kendall's tau since the maximum percentage of people affected is not normally distributed. Both tests were performed considering a 95% level of confidence.

The response capacity of a municipality is determined by its economic condition to recover from a disaster and the presence of master plans that guide the city's occupation process, avoiding the occupation of high-risk areas. As such it merges social, demographic and political characteristics into a single variable. The economic capacity of each municipality to recover from disasters was evaluated by the ratio between total losses from 2000 to 2022 (in USD) and the economic growth attained between 2002 and 2020 (hereafter referred to as "loss/EconGrowth"). The economic growth for each municipality was calculated as the difference in Gross Domestic Product (GDP) between 2020 and 2002. It was not possible to calculate the economic growth between the same years as the losses (2000–2022) due to the availability of GDP data, which only spans from 2002 to 2020[55,56].

We further examined the relationship between the number of climate disasters and the social and demographic characteristics of the population (SPI and Indigenous People) across different classes of municipality population size. Municipalities were grouped into four population classes: ≤20,000; 20,001–50,000; 50,001–300,000; and >300,000 inhabitants. These thresholds are commonly used by Brazil's national statistics agency (IBGE)39 and the Institute for Applied Economic Research (IPEA)55 to distinguish municipal size and capacity, and are also reflected in national policies, including the City Statute (Law 10.257/2001). This law requires municipalities with over 20,000 inhabitants to develop a Master Plan. The master plan guides urban management regarding the city's development, indicating risk areas, such as flooding and landslide risk areas that should not be occupied.

**Compound risk**

To classify municipalities according to their multidimensional compound risk, we employed a dispersion diagram that visually represents the interplay between hazard, exposure, and response capacity. In this diagram, the X-axis displays the maximum percentage of the population affected between 2000 and 2022, capturing both observed hazards and population exposure. The Y-axis represents economic response capacity, quantified as the ratio of total disaster-related losses to economic growth (Loss/EconGrowth, see section 3.1). Municipalities are depicted as colored dots, with each color corresponding to a class of population size (see section 3.2). In this visualization, municipalities with high population exposure and low economic response capacity—indicating those most severely impacted—are concentrated in the upper right quadrant. The distribution of colors within this quadrant provides further insights into the prevalence of master plans among these municipalities.

**Inclusion and ethics statement**

This study relies exclusively on publicly available secondary data and did not involve human subjects, interviews, or identifiable personal information; therefore, ethical approval was not required. The research was developed with a commitment to inclusion and equity, aiming to expose how climate-related disasters disproportionately affect relatively poor, Indigenous, and traditional populations in the Brazilian Amazon. By centering these impacts, the study seeks to inform rights-based and socially just adaptation strategies.

**Reporting summary**

Further information on research design is available in the Nature Portfolio Reporting Summary linked to this article.

## Data availability

All data analyzed or generated during this study are available within the paper. "Supplementary Data 1 and 2" are provided as Excel spreadsheets. The datasets used are publicly available and can be accessed from the sources cited in the manuscript.

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

## Acknowledgements

We acknowledge support from the Climate and Land Use Alliance and the Forest, People and Climate Initiative (CLUA; grant number G-2211-58697), the Mott Foundation (grant number G-2022-10535), and the Moore Destination Initiative of the Gordon and Betty Moore Foundation (grant number 12119). We thank our partners and colleagues across the

Amazon and internationally who contributed to interdisciplinary discussions on the social dimensions of climate risk, especially in the context of Indigenous and traditional communities. Their insights and engagement were instrumental in shaping the perspectives advanced in this study.

## Author contributions

All authors contributed to the writing, discussed the results, and approved the final manuscript. P.F.P. led the conceptual framing, coordination, and writing of the manuscript. R.S., A.A., and L.P. C.G. developed the data analysis, methods and figures and graphs. M.F. and P.M. contributed with regional expertise, including climate-forest dynamics. D.L. provided input on the analytical approach and contributed to the Methods section. L.S. supported the analytical framing and the positioning of the study within broader climate policy and adaptation contexts.

## Competing interests

The authors declare no competing interests.
