## [Transparent Peer Review File · Nature Communications]

Vulnerabilities and compound risks of escalating climate disasters in the Brazilian Amazon

Corresponding Author: Dr Patricia Pinho

Version 0:

Reviewer comments:

Reviewer #1

(Remarks to the Author)

The article produced a consistent and updated compilation of data available in the Atlas of Disasters in Brazil. Although it does not have great scientific depth, the analyses of the dynamic spatial/temporal mapping are robust in evidencing a significant intensification of disasters resulting from more intense and more frequent meteorological and climate extremes throughout the Amazon territory, during the last decades. The methodology used contemplates the state of the art in the integrated assessment of the climate hazards superimposed on the components of exposure and vulnerability (encompassing indicators in the social, economic, and environmental dimensions). The results and information are enlightening and relevant regarding the urgent need to implement public policies for mitigation and adaptation in the face of the threat of compound climate extremes, especially for the Amazon population living in small municipalities and indigenous areas. Given the current major problems of this topic and, considering that the writing and graphic results are in a very communicative and direct scientific format, I am in favor of accepting the article.

The authors should check or review the points below that need improvement in the presentation and reading of the article.

The authors mention "Brazilian Amazon" and "Brazilian Legal Amazon". Check and standardize the name of the study area. If you use an acronym (BRAMZ), it might make it easier to read.

In the Results, the authors make comparative analyses between short-term periods, 2000-2005, 2006-2010, and 2018/2022. Why were 5-year samples chosen?

In the case of decadal samples, e.g., 2000-2009 to 2012/2022, would the results be different?

This relevant methodological aspect should be made clearer in the manuscript.

Fig. 5 depicts the compound risk across Amazonian municipalities very well and the analysis text is excellent. However, no aspect of hydrology is addressed. Therefore, if the plotting of rivers on the map were suppressed (Fig. 5b), the visualization of the symbols would be much better.

Line 280:

"The spatiotemporal exposure of municipalities and populations in the Amazon is closely linked to"
Here references 25 and 26 were cited that did not address studies for the Amazon, is that correct?

Line 308:

Check Citation 33–353

Line 385:

hazel? Is this correct?

Line 432, and line 437:

Check citations 50 for SPI and 48 for the Indigenous population.

Line 457

Reference 51 is missing.

Reviewer #2

(Remarks to the Author)

The paper assesses the exposure and vulnerability of people to five natural disasters—floods, droughts, heatwaves, fires, and landslides—in the Brazilian Amazon over a 22-year period (2000–2022). Conducting analyses at the municipal level, the authors find that smaller municipalities and indigenous populations are disproportionately affected, experiencing significant economic and non-economic losses and damages.

Several methodological aspects could benefit from additional clarification, provided suitable data are available; if not, the authors should explicitly acknowledge these potential limitations. With that said, I believe the findings are of general interest and relevance to a broad audience across social science disciplines and the policy community. Below, I provide my detailed comments, organized into major and minor points, which I hope will help the author(s) strengthen the manuscript.

Major comments:

- Vulnerability dimension: While the results themselves were not entirely surprising—given the known fact that the marginalized, economically disadvantaged, and minority groups often bear the greatest impacts of climate-related shocks—the paper provides valuable quantitative evidence of this pattern in the specific context of the Brazilian Amazon. However, the dimension of vulnerability remains somewhat unclear. To further enhance their contribution, the authors could more explicitly identify and detail specific channels through which these impacts occur (see Hallegatte et al. [2017]; Birkmann et al. [2022]; Triyana et al. [2024]). As currently, the dimension of vulnerability is somewhat of a black box.

1. While the paper effectively identifies the populations most severely affected, it does not provide an in-depth exploration of the underlying factors contributing to their vulnerability. If data availability permits, the authors might consider analyzing sectoral contributions to municipal GDP by population size or providing a detailed breakdown of economic damages by sector (e.g., agriculture, livestock, infrastructure, housing, and healthcare) at the municipal level. Additionally, given that indigenous groups are often disproportionately vulnerable, a more detailed analysis of their demographic dynamics—such as changes over time in the share of Indigenous households across municipalities or the proportion engaged in climate-sensitive livelihoods—would substantially enhance the policy relevance and practical implications of the findings. Nevertheless, there is the issue of reverse causality since natural disaster affect the most vulnerable, but the most vulnerable people are also more susceptible to these shocks.

2. It is also important to recognize that relying solely on economic damage expressed in absolute terms may obscure the true severity of impacts experienced by poorer and marginalized communities. Even if the total monetary value of damages appears small, the relative impact on these households or municipalities can be disproportionately large, as losses represent a greater share of their limited income or assets. These groups typically lack the resources, infrastructure, and social safety nets needed to effectively respond to or recover from shocks. Thus, even modest economic losses can trigger significant setbacks in their livelihoods, health outcomes, and overall resilience. The paper addresses some of this concern by considering the Social Progress Index scores, but explicitly acknowledging and discussing this relative vulnerability further—beyond absolute economic damages—could provide a more comprehensive and accurate understanding of the true implications of climate-related shocks for these communities.

- Hazards:

1. It would be beneficial to examine how multiple shocks overlap or interact. Climate shocks rarely occur in isolation, and simultaneous or sequential events may significantly alter vulnerability and exposure patterns. Analyzing these compounded or overlapping impacts would add substantial value to the study.

2. Although the manuscript states (in lines 263–264) that the study reveals an increase in both the “frequency and intensity” of hazards, I did not find a clear analysis or explicit discussion of changes in hazard intensity. If the authors are implicitly suggesting that increased intensity is reflected by greater numbers of exposed people and higher economic losses, this point should be clarified explicitly. Regardless, further distinguishing between the intensity and frequency of shocks would greatly enhance the interpretation of the results. While frequency is undoubtedly important, differentiating between numerous low-intensity events versus fewer but higher-intensity shocks can provide crucial insights for assessing future risks and guiding targeted adaptation responses.

3. I also wonder if the authors could discuss whether some of the observed increases in exposure might be attributed to improvements in data quality or reporting practices, rather than reflecting only actual increases in hazard exposure.

Minor comments

- The paper seems to use “wet events” and “floods” interchangeably. Greater consistency or explicit clarification of terms would improve readability and ensure the accuracy of interpretations.

- Regarding the classification of municipalities into four size-based categories: Is this classification officially recognized or supported by previous literature? If not, I recommend performing a robustness check of these thresholds to ensure validity.

- The abstract could be improved by clearly distinguishing the authors' original contributions from general descriptive information.

References

- Birkmann, J., Liwenga, E., Pandey R., Boyd, E., Djalante, R., Gemenne, F., Leal Filho, W., Pinho, P. F., Stringer, L., & Wrathall, D. (2022). Poverty, livelihoods, and sustainable development. In H.-O. Pörtner, D. C. Roberts, M. Tignor, E. S. Poloczanska, K. Mintenbeck, A. Alegría, M. Craig, S. Langsdorf, S. Löschke, V. Möller, A. Okem, & B. Rama (Eds.), *Climate change 2022: Impacts, adaptation, and vulnerability. Contribution of Working Group II to the Sixth Assessment Report of the Intergovernmental Panel on Climate Change*, pp. 1171–1274. Cambridge University Press.
- Hallegatte, S., & Rozenberg, J. (2017). Climate change through a poverty lens. *Nature Climate Change*, 7, 250–256.
- Triyana, M., Jiang, A. W., Hu, Y., & Naoaj, M. S. (2024). Climate shocks and the poor: A review of the literature (Policy Research Working Paper 10742). World Bank.

Version 1:

Reviewer comments:

Reviewer #1

(Remarks to the Author)

I am grateful for the opportunity to review the manuscript entitled "Escalating Climate Disasters in the Amazon: Vulnerabilities and Compound Risks." I have thoroughly evaluated the revised version of the manuscript alongside the authors' detailed point-by-point response to my previous comments and recommendations.

The authors have provided comprehensive responses, addressing all my initial concerns with considerable care and scientific rigor. The undertaken revisions are substantial and represent a significant improvement to the manuscript, far exceeding mere superficial edits. The incorporation of additional data and refined analyses, as suggested, has notably strengthened the evidential support and interpretation of the results. Furthermore, the authors have performed extensive textual revisions to rectify ambiguities and incorrect citations and have refined the discussion, which is now markedly more comprehensive and robust. These enhancements have collectively elevated the manuscript's overall clarity, scholarly depth, and academic impact.

The revised manuscript now presents a compelling and well-supported narrative on the robust intensification of climate disasters in the Amazon and their socio-environmental implications during present climate (2000 to 2022). My sole remaining observation pertains to the concluding section, which, in its current form, focuses primarily on the retrospective analysis. To further strengthen the manuscript's impact, a brief forward-looking perspective on the projected increase in frequency and intensity of such climate extreme events under future climate change scenarios would be highly valuable. For instance, integrating a succinct commentary on these worrying future threats, perhaps by referencing relevant works such as de Souza et al. (2024, *Climate*, <https://doi.org/10.3390/cli12070095>), would effectively bridge the authors' robust findings on recent trends with the critical and urgent risks anticipated in the coming decades.

Aside from this minor suggestion for the conclusion, I am fully satisfied with the revisions and have no further substantive considerations. The manuscript now meets the high standards required for publication. Therefore, it is my final decision to recommend acceptance.

Reviewer #2

(Remarks to the Author)

Thank you for carefully revising the paper and for addressing my comments. I appreciate the effort you have made to strengthen the discussion, and I acknowledge that many of my earlier suggestions extend beyond the immediate scope of this paper. It is, however, very useful to see them reflected as limitations or as potential directions for future research.

The paper has improved considerably, particularly in the abstract and discussion sections, which now read more clearly and with a smoother flow. These revisions enhance the overall coherence of the argument and make the contribution of the paper more compelling.

Response to Reviewers

We sincerely thank the reviewers and editors for their thoughtful comments, which have substantially improved the manuscript. Below, we provide detailed responses to each comment. Reviewer comments are shown in *italics*, followed by our responses.

Reviewer #1

The article produced a consistent and updated compilation of data available in the Atlas of Disasters in Brazil. Although it does not have great scientific depth, the analyses of the dynamic spatial/temporal mapping are robust in evidencing a significant intensification of disasters resulting from more intense and more frequent meteorological and climate extremes throughout the Amazon territory, during the last decades.

Response:

Thank you very much for the thoughtful review and for recognizing the relevance, robustness, and clarity of the article.

The authors mention “Brazilian Amazon” and “Brazilian Legal Amazon”. Check and standardize the name of the study area. If you use an acronym (BRAMZ), it might make it easier to read.

Response:

We agree on the importance of consistency when referring to the study area and have revised the manuscript to standardize the terminology, consistently using "Brazilian Amazon." We considered the acronym BRAMZ but prioritized accessibility for a broader readership.

In the Results, the authors make comparative analyses between short-term periods, 2000-2005, 2006-2010, and 2018/2022. Why were 5-year samples chosen? In the case of decadal samples, e.g., 2000-2009 to 2012/2022, would the results be different?

Response:

We selected 5-year intervals to provide a detailed view of medium-term trends while smoothing annual variability. That said, the increasing trend is consistent across the entire time series, as shown in the figure below that is not in the article but suggests that the results remain robust regardless of the interval chosen. We appreciate the suggestion to consider 10-year periods (e.g., 2000–2009 and 2012–2022), and agree that this could offer a complementary perspective. We have now clarified our rationale in the Methods section in track change (page 9) and added a brief reflection in the Discussion (page 7), noting that future studies may benefit from multi-scale temporal analyses.

Trend Consistency in Climate Disasters Across Different Time Intervals

Fig. 5 depicts the compound risk across Amazonian municipalities very well and the analysis text is excellent. However, no aspect of hydrology is addressed. Therefore, if the plotting of rivers on the map were suppressed (Fig. 5b), the visualization of the symbols would be much better

Response:

While we understand the suggestion to suppress rivers for symbol clarity, we chose to retain them. We have made the link between rivers and hydrological exposure more explicit in the Discussion, highlighting rivers’ critical role in shaping connectivity, isolation, and disaster risk.

Line 280: Here references 25 and 26 were cited that did not address studies for the Amazon, is that correct?

Response:

We corrected the references in the text, now citing Brondízio et al. (2016) and Parry et al. (2017), both focusing on the Amazon.

Line 308: Check Citation 33–35.

Response:

We have corrected these citations to Carr-Hill (2013) and Galappaththi et al. (2020).

Line 385: hazel? Is this correct?

Response:

We corrected this to "haze," referring to reduced visibility from biomass burning.

Line 432 and 437: Check citations 50 for SPI and 48 for the Indigenous population.

Response:

We corrected the citations. SPI is now properly cited from Santos et al. (2021), and Indigenous population estimates from IBGE (2023).

Line 457: Reference 51 is missing.

Response:

We added Reference 51: IPEA exchange rate data, now properly cited in the manuscript.

Reviewer #2

Several methodological aspects could benefit from additional clarification, provided suitable data are available; if not, the authors should explicitly acknowledge these potential limitations.

Response:

Where possible, we have clarified our methods (page 8, 9, 10) and explicitly acknowledged limitations and suggested directions for future research (page 7).

Vulnerability dimension: While the results themselves were not entirely surprising, the paper provides valuable quantitative evidence. However, the dimension of vulnerability remains somewhat unclear. Authors could more explicitly identify and detail specific channels through which these impacts occur.

Response:

We appreciate this insightful comment. In response, we have expanded the Discussion to more clearly articulate the channels through which climate-related shocks deepen vulnerability in the Amazon. Drawing on Hallegatte et al. (2017), Birkmann et al. (2022), and Triyana et al. (2024), we highlight factors such as the invisibility of Indigenous and traditional livelihoods in official data systems, their concentration in high-risk zones, and their dependence on climate-sensitive sectors like farming. These structural conditions contribute to disproportionate and often underreported impacts, reinforcing long-term socioeconomic vulnerability (see revised text on page 7).

While the paper effectively identifies the populations most severely affected, it does not provide an in-depth exploration of the underlying factors contributing to their vulnerability. If data availability permits, the authors might consider analyzing sectoral contributions to municipal GDP by population size or providing a detailed breakdown of economic damages by sector (e.g., agriculture, livestock, infrastructure, housing, and healthcare) at the municipal level. Additionally, given that indigenous groups are often disproportionately vulnerable, a more detailed analysis of their demographic dynamics—such as changes over time in the share of Indigenous households across municipalities or the proportion engaged in climate-sensitive livelihoods—would substantially enhance the policy relevance and practical implications of the findings. Nevertheless, there is the issue of reverse causality since natural disaster affect the most vulnerable, but the most vulnerable people are also more susceptible to these shocks.

Response:

We have calculated the sectoral breakdown of economic damages for the nine high-risk municipalities shown in Figure 5. Farming accounts for the largest share (~50%), followed by infrastructure and housing. As this pattern is consistent with the overall trends reported for the

entire region, we opted not to include this breakdown in the main text. While limitations in publicly available data currently prevent us from assessing changes over time in the share of Indigenous households across municipalities or the proportion engaged in climate-sensitive livelihoods, we fully recognize the value of such an analysis and appreciate the suggestion. We now acknowledge this limitation explicitly in the discussion and highlight it as a critical gap for future research.

It is also important to recognize that relying solely on economic damage expressed in absolute terms may obscure the true severity of impacts experienced by poorer and marginalized communities. Even if the total monetary value of damages appears small, the relative impact on these households or municipalities can be disproportionately large, as losses represent a greater share of their limited income or assets. These groups typically lack the resources, infrastructure, and social safety nets needed to effectively respond to or recover from shocks. Thus, even modest economic losses can trigger significant setbacks in their livelihoods, health outcomes, and overall resilience. The paper addresses some of this concern by considering the Social Progress Index scores, but explicitly acknowledging and discussing this relative vulnerability further—beyond absolute economic damages—could provide a more comprehensive and accurate understanding of the true implications of climate-related shocks for these communities.

Response:

While our analysis addresses this issue by comparing economic losses to municipal economic growth (Fig. 5a) and incorporating Social Progress Index (SPI) scores to reflect structural vulnerabilities (Section 5), we have now explicitly emphasized it in the Discussion (page 7) as well. We highlight how relative losses disproportionately affect resilience and recovery in marginalized populations, often culminating in existential threats that are not fully captured by conventional economic metrics.

Hazards: It would be beneficial to examine how multiple shocks overlap or interact.

Response:

We agree that understanding how multiple shocks interact, whether simultaneously or sequentially is crucial for assessing compounding risks and their impacts on vulnerability and exposure. However, a full-scale analysis of overlapping and interacting shocks requires additional methodological development and disaggregated data, which are beyond the scope of the current

paper and which would merit a paper of their own to enable rich discussion. Further research is underway to explore these interactions and their dynamics in greater depth.

Although the manuscript states that the study reveals an increase in both the “frequency and intensity” of hazards, I did not find a clear analysis of changes in hazard intensity.

Response:

We agree that clarifying the distinction between frequency and intensity is critical to improving the interpretation of hazard dynamics and their implications for adaptation planning. We don't assess intensity per se, and instead infer from the literature the duration, and or the recurrence (consecutive years at the municipality level) of high-impact events, and their extended disruptions, e.g., of prolonged droughts. Our use of the literature for this purpose is now made clear in the discussion section (6).

I also wonder if the authors could discuss whether some of the observed increases in exposure might be attributed to improvements in data quality or reporting practices.

Response:

We acknowledge that improvements in reporting practices may contribute to the observed increase in recorded events. However, studies, such as Marengo et al. (2024), Espinoza et al. (2024) and Alencar (2022), document a clear intensification of extreme events in the Amazon, including more frequent and severe droughts, floods and fires. These findings support our interpretation that the trends observed in our study reflect actual increases in hazard frequency and intensity, rather than solely improved reporting (page 6).

Minor comment: The paper seems to use “wet events” and “floods” interchangeably.

Response:

We have now clarified throughout the manuscript that "wet events" refer broadly to floods and intense rainfall, while "floods" are used specifically where appropriate.

Minor comment: Regarding the classification of municipalities into four size-based categories.

Response:

We now clarify in the Methods that our classification is aligned with IBGE and IPEA practices and public policies in Brazil, and we added citations.

Minor comment: The abstract could be improved by clearly distinguishing the authors' original contributions from general descriptive information.

Response:

We have revised the abstract to clearly emphasize our original contributions, including the scale of population exposure, disaster frequency increases, economic loss distribution, and disproportionate impacts on smaller Indigenous-majority municipalities.